# Food disgust sensitivity predicts disease-preventing behaviour beyond the food domain in the COVID-19 pandemic in Germany

**Jeanine Ammann**◉*, **Meret Casagrande**

Department of Health Science and Technology (D-HEST), Consumer Behaviour, ETH Zurich, Zurich, Switzerland

* jeanine.ammann@alumni.ethz.ch

**Data Availability Statement:** The data underlying the results presented in the study are available from https://doi.org/10.3929/ethz-b-000490138.

## Abstract

In the fight against the COVID-19 pandemic, personal hygiene behaviours such as proper handwashing have gained significantly more attention and interpersonal contact is performed with great care. Disgust, as a disease-avoidance mechanism, can play an important role in the promotion of hygiene behaviour. We know from previous research that pathogen disgust can be a predictor of an individual's behaviour in the pandemic. Given that the pandemic greatly affects our food and eating behaviour, the current study aims to add to the existing evidence and to complement it by investigating the role of food-specific disgust in the pandemic. For that, we conducted an online survey in Germany in April 2020, while the pandemic was spreading in Europe. A total of 519 participants completed the survey and provided information about their COVID-19-related attitudes and behaviours and about their food disgust sensitivity. The results show that food disgust sensitivity is an important predictor for an individual's feelings, shopping behaviour, and disease-preventive behaviour related to the COVID-19 pandemic. Given that the success of political measures to fight the pandemic critically depends on the population to support and follow the proposed measures, a better understanding of the factors driving individual behaviour is key. Implications for pandemic management are discussed.

## 1 Introduction

In times of a global health crisis, quick mitigation is of crucial importance to weaken the impact the crisis has on people's health and wealth. Besides governmental action aiming to fight the crisis, public support is key for the success of these actions. Both demographic and psychological factors influence an individual's compliance. The current paper investigates what role sociodemographic variables and the psychological factor food disgust, which can be seen as a disease-avoidance mechanism, play in the successful mitigation of a pandemic.

**Funding:** The authors received no specific funding for this work.

**Competing interests:** The authors have declared that no competing interests exist.

## 1.1 COVID-19

In December 2019, the first cases of infection with the novel coronavirus, COVID-19, were reported in Wuhan, China, from where the virus spread rapidly across the globe [1]. In March 2020, the virus has developed into a pandemic and by July 2020, more than 12 million cases of COVID-19 have been reported [2, 3]. Symptoms caused by the virus include fever, cough, fatigue, and pneumonia [4, 5]. Older patients have a significantly higher risk of severe or fatal progression of the infection than younger people [1, 6]. Importantly, infected individuals can also appear asymptomatic and thus unknowingly help the transmission of the virus [7].

## 1.2 Mitigating the spread of COVID-19

To contain the spread of the virus form one person to another, the World Health Organisation (WHO) has proposed various measures including regular and proper handwashing with soap, wearing masks, and physical distancing [8, 9]. Adopting these behaviours helps reduce or delay the local spread of the virus, aims to avoid an overloading of the healthcare system, and to protect vulnerable groups [3, 10]. Countries differ widely in their attempts to mitigate the spread of the virus and in the measures they choose to do so [3]. Individuals, on the other side, differ in the extent to which they follow these government recommendations and whether and how they take health precautions. The best measures are worthless if the public is unwilling to comply [11]. With that, individual behaviour can be decisive for the success of the measures aiming to control the spread of the virus [3].

## 1.3 Motivation for compliance with virus-mitigating behaviour

Various factors influence whether individuals comply with behaviour recommendations or not. Besides sociodemographic information such as sex [12] and risk perception [13], personal abilities, people's intrinsic motivation and individual differences in psychological variables were found to play an important role in whether they follow virus-mitigating behaviour recommendations [14, 15]. One of these psychological factors is disgust [16]. It is part of the behavioural immune system, a proactive system, which aims to mitigate the threat of disease before infection occurs [16, 17]. Tybur, Lieberman [18] identified three domains of disgust: (1) pathogen disgust, (2) sexual disgust and (3) moral disgust. Pathogen disgust can be seen as reactiveness to contamination threats and motivates avoidance of infectious microorganisms, resulting in reduced pathogen contact and lower risk of infection [19–21]. Minimising exposure to pathogens, disgust also impacts our hygiene, sexual, and social behaviour [22, 23]. Social behaviour is especially important in a pandemic, when physical distancing is recommended [24].

## 1.4 Disgust and virus-mitigating behaviour

We know from previous research, that pathogen disgust correlates with food disgust sensitivity [25]. However, food disgust covers more than the avoidance of pathogens [26]. It also includes cues that are neither pathogen-related nor pose a health risk. Nevertheless, food disgust can play a crucial role in hygiene behaviour. For instance, food disgust sensitivity was identified as an important driver for hygiene behaviour in the home kitchen [27]. Individuals with higher levels of food disgust sensitivity reported to perform hygiene behaviour more frequently than individuals with lower levels of food disgust sensitivity, thereby maintaining higher levels of food safety in their home kitchen. Similarly, disgust sensitive individuals were more reluctant to consume food that has passed its expiration date as compared to less disgust sensitive individuals [27]. Disgust has further been identified as important motivator for handwashing [28, 29], the

frequency of cleaning or disinfecting [30], and disgust-based cues have successfully been used to promote handwashing [31]. Given this evidence, it is not surprising that disgust was also found to play an important role in the context of COVID-19, where hygiene behaviour and pathogen avoidance are crucial to contain the spread of the virus. Specifically, participants' behaviour in terms of hygiene and physical distancing was predicted by pathogen disgust [32].

## 1.5 Aim

It has been argued that without the hygiene behaviour motivated by disgust, infectious diseases would cause far higher mortality [33]. Furthermore, mortality and disease threats have been associated with elevated levels of disgust [17]. Given that disgust helps disease avoidance and has been linked to individuals' intention to comply with behaviour recommendations in the context of COVID-19, the present study explored the role of food disgust sensitivity as predictor for attitudes and behaviours related to the current pandemic. We chose a food-specific measure of disgust sensitivity for three reasons. First, it has been demonstrated that there is a relationship between pathogen disgust and behaviours related to the pandemic [32]. Given that we know that pathogen disgust and food disgust sensitivity are positively correlated [25], we aimed to add to the evidence by investigating the role food disgust sensitivity plays in mitigating the effects of the pandemic. Second, the current pandemic has a huge impact on how we buy, prepare, and consume food. Therefore, we expect some conceptual overlap between food disgust sensitivity, hygiene, and behaviours related to the pandemic. Third, we aim to investigate whether food disgust sensitivity, due to its risk avoidance nature, explains variance beyond the domain of food. Given that enforcement of virus-mitigating measures is costly and public compliance cannot be taken for granted [11], a better understanding of the mechanisms and motivations underlying virus-mitigating behaviours and attitudes can crucially contribute to the success of political measures aiming to contain the spread of the virus.

## 2 Methods

### 2.1 Participants

In April 2020, we conducted an online study with participants from Germany. We used an internet panel obtained from a commercial, ISO-certified panel provider (Respondi AG) to recruit participants. Equal numbers of male and female participants were obtained by applying quotas. We aimed for at least 250 male and female participants each. Assuming small to medium effect sizes (d = .25), this sample size provides ample power (.80) to detect possible effects ($\alpha$ = .05) with a one-tailed test [34]. Similarly, a sample of 500 provides sufficient power to compare dependent Pearson's correlations with common index [35].

Completion of the online survey took between 10 and 15 minutes. Before starting with the survey, participants provided their written consent. This study was approved by the Ethics Committee of ETH Zurich (application 2020-N-47). The survey was completed by a total of 538 participants, of which we excluded 19 participants because the time they took to complete the survey was less than half the median of the survey duration calculated for the whole sample [for example, 36]. As a result, the final sample consisted of 519 participants (50% females). Participants' age ranged from 21 to 71 years (*M* = 50, *SD* = 13).

### 2.2 Questionnaire

The final questionnaire consisted of three sections. In the first section, we asked participants about demographic and individual data, such as participants' age, sex, educational level, and dietary habits, for example, whether they were vegan or vegetarian. This section also included

questions about who was mainly responsible for grocery shopping in the household, how often they used to cook before the pandemic, and how often they cooked at the time of data collection. Furthermore, we asked participants to indicate whether they considered themselves as part of the group of people with higher risk from COVID-19.

Given that previous research reported a relationship between risk perception and behaviour related to the pandemic [13], we investigated self-reported behaviour and feelings regarding the COVID-19 pandemic in the second section (*COVID-19 items*). We asked participants to indicate their level of agreement with 24 items. When selecting the items, we aimed to cover the three domains of *feelings and fears*, *preventive and protective behaviour*, and *shopping and stocking up*. The domain *feelings and fears* describes is a measure for how and how much individuals feel threatened or affected by the virus (e.g., I am afraid of contracting the virus). Items selected for the domain *preventive and protective behaviour* are mostly behaviours which are commonly recommended to help contain the virus (e.g., washing hands with soap). Finally, the domain *shopping and stocking up* describes shopping behaviours that either help fight the virus (e.g., buying disinfectant) or which help reduce exposure to the virus (e.g., stock buying to reduce trips to the store). Responses for all items were given on a six-point scale from 1 (*do not agree at all*) to 6 (*agree very much*). The complete list of selected items can be found in the S1 Appendix.

The third and final section included a measure of disgust sensitivity. To the best of our knowledge, the FDS short is the only available disgust measure that specifically focuses on the domain of food. As a result, we used the 8-item short version of the Food Disgust Scale [FDS short, 26] as a measure of food-specific disgust. The FDS short includes eight food-specific items from different domains of food disgust (meat, mould, fruit, fish, hygiene, vegetable, human contamination, and living contamination). Participants rated these eight situations or products on a scale from 1 (*not disgusting at all*) to 6 (*extremely disgusting*). Sample items were *"To eat with dirty silverware in a restaurant"* or *"To eat brown-coloured avocado pulp"*. The scale had good reliability (8 items, $\alpha = .79$, $M = 4.11$, $SD = 1.00$).

## 2.3 Data analysis

We conducted principal component analysis (PCA) to confirm the assumed one-dimensional structure of the three domains *feelings and fears*, *preventive and protective behaviour*, and *shopping and stocking up* which we used to measure participants' self-reported disease-preventive behaviour. We considered factors with eigenvalues larger than one as relevant and made use of the interpretability criterion, that is, we made sure that all factors were interpretable. We used the Kaiser-Meyer-Olkin measure ($KMO \geq .5$) and Bartlett's test of sphericity ($p < .05$) to determine the adequacy of the dataset for a factor analysis [37]. We calculated Cronbach's alpha coefficients to check the reliability and internal consistencies of the new scales (values $>$ .6 were considered adequate). In addition, we investigated the relationship between the factors and food disgust sensitivity using Pearson's correlations, and finally, we assessed the influence of food disgust sensitivity on disease-preventive behaviour using multiple hierarchical regression models. We analysed all data with Statistical Package for the Social Sciences (SPSS) version 25 (IBM, New York, USA) for Windows. Power calculations to compute the required sample size were done using G*Power version 3.1.9.4 for Windows [34].

## 3 Results

### 3.1 Participants

About half of our sample (52%) indicated not to belong to the group of people with higher risk from COVID-19, 38% believed themselves to be at a higher risk, and 10% did not know. Only 1% of the sample stated that they had already contracted COVID-19. Asking participants how

frequently they used to cook meals at home before the pandemic and at the time of data collection revealed that the self-reported cooking frequency during the pandemic was higher than before the pandemic ($t = 5.93$, $p < .001$).

## 3.2 Disease-preventing behaviour and fears related to COVID-19

Reliability analyses revealed that one of the *feelings and fears* items was problematic (see Table 4 in S1 Appendix for the complete list of items). We ran a PCA on the remaining six items. The results confirmed the one-dimensional structure. The overall KMO measure was .84 and Bartlett's test of sphericity was statistically significant ($p < .001$). Three of the *preventive and protective behaviour* had to be excluded on the basis of the reliability analyses. We ran a PCA on the remaining seven items. The results confirmed the one-dimensional structure. The overall KMO measure was .82 and Bartlett's test of sphericity was statistically significant ($p < .001$). Reliability analyses revealed that one of the *shopping and stocking up* items was problematic. We ran a PCA on the remaining 6 items. The results confirmed the one-dimensional structure. The overall KMO measure was .78 and Bartlett's test of sphericity was statistically significant ($p < .001$). For each of the three factors, we computed an averaged rating scale. All scales had good reliabilities ($\alpha < .75$). The final list of items, including mean values, reliabilities, and factor loadings, can be found in Table 1.

**Table 1. Items used to assess attitudes and behaviours related to the COVID-19 pandemic, including their factor loadings, means (M), 95% confidence intervals (CI), and standard deviations (SD), N = 519.**

| # | Item | Factor loading | *M* | *SD* |
|---|---|---|---|---|
| | Feelings and fears (6 items, $\alpha = 0.87$) | | | |
| 20 | I am afraid of contracting the coronavirus. | .858 | 3.75 | 1.75 |
| 21 | I am afraid of contracting my family and friends with the coronavirus. | .821 | 4.30 | 1.71 |
| 2 | I feel threatened by the current situation regarding the spread of the coronavirus in Germany. | .795 | 3.62 | 1.59 |
| 22 | I am afraid of contracting strangers with the coronavirus. | .785 | 3.91 | 1.68 |
| 23 | I am afraid of an overloading of the health care system. | .757 | 3.91 | 1.68 |
| 1 | I feel affected by the current situation regarding the spread of the coronavirus in Germany. | .673 | 4.28 | 1.53 |
| | Preventive and protective behaviour (7 items, $\alpha = 0.77$) | | | |
| 17 | I keep distance when I meet or come across people. | .784 | 5.44 | 1.02 |
| 8 | I avoid shaking hands with other people. | .715 | 5.49 | 1.09 |
| 6 | I always wash my hands thoroughly with soap. | .684 | 5.34 | 1.04 |
| 4 | I consciously try not to touch my face with my hands. | .672 | 4.56 | 1.48 |
| 19 | When I develop symptoms like fever or cough, I stay at home. | .634 | 5.51 | 1.04 |
| 18 | I try not to touch door handles any more. | .620 | 4.22 | 1.73 |
| 14 | I try to avoid public transport. | .560 | 4.99 | 1.66 |
| | Shopping and stocking up (6 items, $\alpha = 0.77$) | | | |
| 12 | I am increasingly buying long-life food (e.g. canned food, pasta, . . .). | .813 | 3.04 | 1.68 |
| 5 | I have built up an emergency supply of food. | .771 | 2.97 | 1.67 |
| 7 | I have built up an emergency supply of toilet paper. | .754 | 2.44 | 1.64 |
| 10 | I am increasingly buying packed food (e.g. packed vegetables instead of vegetables in open sale). | .682 | 2.61 | 1.75 |
| 9 | I bought disinfectant. | .599 | 3.68 | 2.08 |
| 24 | I am afraid of no longer being able to get the products of daily use (food, toilet paper, . . .). | .507 | 2.79 | 1.65 |

*Note*. Statements were rated on a scale from 1 (*do not agree at all*) to 6 (*completely agree*).

**Table 2. Pearson's correlations for food disgust sensitivity, attitudes or behaviours related to the COVID-19 pandemic, age, and sex N = 519.**

| | 1 | 2 | 3 | 4 | 5 | 6 |
|---|---|---|---|---|---|---|
| 1. FDS short | 1 | | | | | |
| 2. Feelings and fears | .15*** | 1 | | | | |
| 3. Preventive and protective behaviour | .20*** | .46*** | 1 | | | |
| 4. Shopping and stocking up | .27*** | .38*** | .25*** | 1 | | |
| 5. Age | .03 | .10* | .20*** | <-.01 | 1 | |
| 6. Sex | -.16*** | -.16*** | -.25*** | -.07 | -.09* | 1 |

*Note*. FDS short: eight-item Food Disgust Scale [26]; feelings and fears: items related to fears and feelings; preventive and protective behaviour: items related to COVID-19-preventive behaviours, shopping and stocking up: items related to shopping and hoarding behaviours; sex: 0 = females, 1 = males.

*$p < .05$

**$p < .01$

*** $p < .001$.

## 3.3 Disease-preventing behaviour and food disgust sensitivity

Table 2 depicts the relationship between food disgust sensitivity, the COVID-19 items, age, and sex. As widely reported in the literature [for instance, 38, 39], food disgust sensitivity was significantly associated with sex, with females being more disgust sensitive than males. Furthermore, food disgust sensitivity was significantly associated with all three domains *feelings and fears*, *preventive and protective behaviour*, and *shopping and stocking up*. With increasing food disgust sensitivity, participants reported higher levels of fear related to COVID-19, higher frequencies for preventive or protective behaviour and higher frequencies for shopping and stocking behaviour that aims to reduce exposure to the virus.

To investigate the influence of food disgust sensitivity on the coronavirus-related behaviours and feelings while controlling for the effect of age and sex, multiple hierarchical regression analyses were run. The final models predicting COVID-19-related feelings, shopping behaviour, and behaviours aiming to minimise the infection rate were significant and explained 5%, 8%, and 12% of the variance, respectively (Table 3). After controlling for age and sex, food disgust sensitivity was a significant predictor in all three models.

**Table 3. Multiple regression analysis predicting the attitudes or behaviours related to the COVID-19 pandemic from age, sex, and food disgust sensitivity, N = 519.**

| | Feelings and fears | | | Preventive and protective behaviour | | | Shopping and stocking up | | |
|---|---|---|---|---|---|---|---|---|---|
| Variable | B | SE | ß | B | SE | ß | B | SE | ß |
| Step 1 | | | | | | | | | |
| Constant | 3.680*** | .240 | | 4.676*** | .153 | | 3.043*** | .223 | |
| Sex | -.386** | .114 | -.148 | -.393*** | .072 | -.229 | -.158 | .106 | -.066 |
| Age | .008 | .004 | .082 | .012*** | .003 | .179 | -.001 | .004 | -.009 |
| $R^2$ | .03 | | | .09 | | | < .01 | | |
| F | 8.17*** | | | 26.09*** | | | 1.11 | | |
| Step 2 | | | | | | | | | |
| Constant | 2.957*** | .338 | | 4.086*** | .214 | | 1.679*** | .306 | |
| Sex | -.331** | .114 | -.127 | -.348*** | .072 | -.202 | -.053 | .103 | -.022 |
| Age | .008 | .004 | .080 | .012*** | .003 | .177 | -.001 | .004 | -.014 |
| FDS short | .172** | .057 | .131 | .140*** | .036 | .163 | .325*** | .052 | .270 |
| $R^2$ | .05 | | | .12 | | | .08 | | |
| F | 8.55*** | | | 22.92*** | | | 13.97*** | | |

*Note*. Sex: 0 = females, 1 = males

## 4 Discussion

The amount of scientific literature dealing with the COVID-19 pandemic has increased massively, in the medical domain and beyond [40]. To optimise the efforts to mitigate the spread of the virus, it is crucial to understand individuals' motivation to follow behaviour recommendations. So far, relatively few studies have looked at psychological factors predicting behaviours related to the COVID-19 pandemic [14]. On the basis that disgust has been associated with both behaviours related to the pandemic [32], general hygiene behaviour such as hand-washing behaviour [29, 41], and food-specific hygiene behaviour in the home kitchen [27], the present paper investigated the role food disgust sensitivity plays in this major health crisis caused by COVID-19. Specifically, the paper investigated the predictive power of food disgust sensitivity for various behaviours related to the mitigation of the pandemic, which are important to understand the related attitudes and behaviours.

### 4.1 Disgust

Our results indicated that individuals with high food disgust sensitivity tend to express more fear related to the COVID-19 pandemic, perform the recommended behaviours more frequently, and show a higher tendency for hoarding behaviour than individuals with lower food disgust sensitivity. With that, it seems that these individuals maintain higher levels of safety. Fear, as a response to an infection risk, can cause avoidance behaviour and thereby reduce the risk of infection [16]. The virus-mitigating behaviours specifically aim to reduce the risk of infection or spread of the virus, and hoarding behaviour can reduce the necessity to go shopping and thereby reduces possible exposure to the virus. Still, it needs to be acknowledged that correlation does not equal causation. The pandemic comes with an increased pathogen pressure, which in turn can up-regulate disgust sensitivity. For instance, Stevenson, Saluja [42] found that disgust levels assessed during the time of the pandemic were higher than the values reported before the pandemic. Similarly, we noticed that the average food disgust sensitivity in our sample was higher than the values reported in previous studies conducted in Germany before the pandemic [M = 3.84 [females] and M = 3.58 [males]; 25]. These findings may serve as an indication that increased pathogen threat, as imposed through the pandemic, leads to an increase in disgust sensitivity. However, these interpretations should be further underpinned by additional research.

The regression models explained between 5 and 12% of variance in COVID-19-related attitudes and behaviour. Not surprisingly, these values are in a slightly lower range than what previous studies reported for food-related behaviours [27, 43]. Given that the FDS short is a domain-specific disgust measure which focuses on food, this finding was expected. The correlations reported herein between attitudes and behaviours related to COVID-19 and food disgust sensitivity ranged from 0.15 to 0.27, which can be seen as small- to large-sized effects [44].

The highest correlation values were found for the *shopping and stocking up* items. One explanation for this finding certainly stems from the fact that four out of the six shopping items deal with food. Given that these behaviours ultimately aim to reduce the risk of infection with the virus (e.g., by buying products that help prevent an infection), they serve the same risk avoidance purpose as disgust. For the hoarding behaviours, we reason that these behaviours are a means for individuals to deal with the uncertainty and panic caused by the pandemic. Obtaining emergency supplies of food and toilet paper not only provides individuals with a certain degree of power and agency in these uncertain times but also reduces the risk of infection by making sure that basic supplies are available and thereby turning frequent shopping trips unnecessary. Similar effects were found for books, as Nguyen, Tran [45] reported

that due to the pandemic, many consumers no longer go to the stores but avoid taking this health risk by buying the products online instead.

## 4.2 Sociodemographic correlates

In line with previous research [3, 11, 15], we found that females showed more compliance with the COVID-19 behaviour recommendations than males. For instance, Capraro and Barcelo [12] found that females were more likely to wear face coverings than males. Our research shows that this effect also applies to other types of preventive behaviours such as staying at home or keeping some distance. Females also reported higher levels of fear related to the COVID-19 pandemic than males. Given that with older age, individuals face a higher risk of mortality caused by COVID-19, a relationship with age was to be assumed. Indeed, age was a significant predictor for virus-mitigating behaviour. This is further in line with previous research conducted in France [11].

## 4.3 Limitations and outlook

Our study has a few limitations that need to be addressed. Our data relies on self-report, which makes it subject to individual bias. It might be interesting for future research to compare our findings with observational data. Another question that should be addressed henceforth is whether these findings can be translated to other pandemics as well. Given that similar constructs for food disgust sensitivity have been used cross-culturally [25, 46], another interesting endeavour would be to test whether these relationships can be found across different countries. Not only have individuals been affected differently by the pandemic depending on their sociodemographic background [47], but individual disgust sensitivity also depends strongly on the environment in which an individual lives [see 48 for a discussion].

Our research has implications for research and practice. The field of food disgust is still young, and many discoveries remain to be made. Our study adds another piece to the puzzle. In terms of practice, the design of interventions to mitigate the spread of the virus can build on our findings. As the pandemic progresses, it becomes harder to motivate the public to follow the behaviour recommendations. In this regard, social science can play an important role in supporting behaviour change [49]. Based on our results, we suggest that the use of disgust cues can be used as a measure to motivate preventive behaviour. Young, Reimer [50] found that for most safe food handling constructs, there were no consistent relationships between knowledge and behaviour. As argued elsewhere [27], we reason that disgust can trigger an automatic response and thereby help promote disease avoidant behaviour in addition and in support to the promotion of knowledge. The COVID-19 pandemic not only shapes people's behaviour during the pandemic but it is also expected to affect behaviour patterns in the long term [51]. It will be interesting for future research to examine whether the pandemic has led to a change in individual disgust sensitivity.

## 5 Conclusion

The present study demonstrates the important role food disgust can play as disease avoidance mechanism in the context of a global health crisis and beyond the domain of food. Specifically, our results indicate that food disgust sensitivity is a significant predictor for an individual's feelings, shopping behaviour, and disease-preventive behaviour related to the COVID-19 pandemic. With that, our findings can be used to support policy and practice in the design and implementation of measures and recommendations to help contain the spread of a virus. Further research is needed to test these associations in an experimental setting and investigate whether they translate into actual behaviour. Finally, we find that individuals with high food

disgust sensitivity may not only protect themselves from foodborne diseases but may also contribute significantly to the mitigation of the COVID-19 pandemic.

## Supporting information

**S1 Appendix.**
(DOCX)

## Acknowledgments

The authors thank Prof. Dr. Michael Siegrist for his valuable support in the study design and critical feedback on the manuscript.

## Author Contributions

**Conceptualization:** Jeanine Ammann, Meret Casagrande.

**Data curation:** Meret Casagrande.

**Formal analysis:** Meret Casagrande.

**Investigation:** Meret Casagrande.

**Methodology:** Jeanine Ammann, Meret Casagrande.

**Project administration:** Jeanine Ammann.

**Supervision:** Jeanine Ammann.

**Writing – original draft:** Jeanine Ammann.

**Writing – review & editing:** Jeanine Ammann, Meret Casagrande.

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
