## [Decision Letter · Decision Letter 0]

21 May 2021

PONE-D-21-12532

Food disgust sensitivity predicts disease-preventing behaviour beyond the food domain in the COVID-19 pandemic in Germany.

PLOS ONE

Dear Dr. Ammann,

Thank you for submitting your manuscript to PLOS ONE. After careful consideration, we feel that it has merit but does not fully meet PLOS ONE’s publication criteria as it currently stands. Therefore, we invite you to submit a revised version of the manuscript that addresses the points raised during the review process.

Please find below the reviewer's comments, as well as those of mine.

We look forward to receiving your revised manuscript.

Kind regards,

Valerio Capraro

Academic Editor

PLOS ONE

Additional Editor Comments:

I have now collected one review from one expert in the field. I was unable to find a second reviewer, but I am myself familiar with the topic of this manuscript, therefore I feel confident in making a decision with only one review. The reviewer is positive and suggests minor revision. I agree with the reviewer. Therefore, I would like to invite you to revise your work for Plos One. Needless to say that all the reviewer's comments should be addressed. Moreover, I would like to add a few more comments from my own reading of the paper, mainly regarding the literature review. Indeed, you report several interesting results that are in line with previous work, and so they should be discussed in this larger context: (i) You report a correlation between feeling of fears and preventative behaviour. This correlation has been reported also in other studies (Capraro & Barcelo, 2020; Dryhurst et al. 2020); (ii) you find gender differences in preventative behavior. This was also reported previously (Capraro & Barcelo, 2020); (iii) This "perspective article" on what social and behavioural science can do to support pandemic response can be useful: Van Bavel et al. (2020).

I am looking forward for the revision.

Capraro V, Barcelo H (2020) The effect of messaging and gender on intentions to wear a face covering to slow down COVID-19 transmission. Journal of Behavioral Economics for Policy 4, Special Issue 2, 45-55.

Dryhurst, S., Schneider, C. R., Kerr, J., Freeman, A. L., Recchia, G., Van Der Bles, A. M., ... & van der Linden, S. (2020). Risk perceptions of COVID-19 around the world. Journal of Risk Research, 23(7-8), 994-1006.

Van Bavel JJ, et al (2020) Using social and behavioural science to support COVID-19 pandemic response. Nature Human Behaviour 4, 460-471.

Journal Requirements:

4. We note you have included a table to which you do not refer in the text of your manuscript. Please ensure that you refer to Table 3 and 4 in your text; if accepted, production will need this reference to link the reader to the Table.

Reviewers' comments:

Reviewer's Responses to Questions

**Comments to the Author**

1. Is the manuscript technically sound, and do the data support the conclusions?

Reviewer #1: Partly

2. Has the statistical analysis been performed appropriately and rigorously? 

Reviewer #1: Yes

3. Have the authors made all data underlying the findings in their manuscript fully available?

Reviewer #1: No

4. Is the manuscript presented in an intelligible fashion and written in standard English?

Reviewer #1: Yes

5. Review Comments to the Author

Reviewer #1: The study investigated the link between food disgust sensitivity and an individual’s feelings, shopping behavior, and disease-preventive behavior related to the COVID-19 pandemic. The research question is interesting and the data are a valuable contribution to the field. Overall, it is an interesting and well-written manuscript. I enjoyed reading the submission very much. I have only minor comments to further improve the manuscript.

1. My main concern: causation needs to be more carefully addressed. It seems, that authors assumed that increased food disgust sensitivity may predict higher levels of fear related to COVID-19. However, higher pathogen threat (i.e. fear related to COVID-19) may influence disgust sensitivity and pathogen avoidance behavior. It is worth citing some research that demonstrates experimental manipulations of pathogen threat might up-regulate disgust sensitivity (i.e., behavioral immune system processes). The following two papers may (or may not) be useful for some of the discussion on the link between disgust sensitivity and pathogen threat during COVID-19 pandemic: https://doi.org/10.3389/fpsyg.2020.600761
https://doi.org/10.3389/fpsyg.2021.622634

The language throughout needs to be more cautious and authors should be more attentive to other potential causal scenarios.

2. As a general rule, a full stop is not used at the end of a title.

3. In the introduction, Authors write, “A common disgust response is avoidance, resulting in reduced pathogen contact and lower risk of infection (…) disgust is a pathogen-avoidance mechanism….” Consider specifying pathogen disgust here (reactiveness to contamination threats), since sexual and moral disgust do not appear to neutralize infection (see https://doi.org/10.1037/a0030778).

4. Perhaps a little more justification as to why, only food disgust sensitivity was measured. More space should be devoted in the introduction and methods to describing why you chose the measures you chose (the Food Disgust Scale), and why certain related measures (e.g., the Three Domain Disgust Scale, the Disgust Scale-Revised or sets of images associated with contagion risk) were not included. More transparency about your design decisions would be helpful for readers to evaluate the study's methods.

5. Any a priori decisions that factored into your sample size (rules of thumb, power analyses) should reported.

6. In affluent parts of the world it is possible to choose to avoid a pathogen if you think one may be present e.g. in expired food as other meals are available. In some environments people have no way to avoid similar warning signs – can it be assessed in future research directions?

6. PLOS authors have the option to publish the peer review history of their article (what does this mean?). If published, this will include your full peer review and any attached files.

Reviewer #1: No

---

## [Author Response · Author response to Decision Letter 0]

21 Jun 2021

Reviewer #1: 

The study investigated the link between food disgust sensitivity and an individual’s feelings, shopping behavior, and disease-preventive behavior related to the COVID-19 pandemic. The research question is interesting and the data are a valuable contribution to the field. Overall, it is an interesting and well-written manuscript. I enjoyed reading the submission very much. I have only minor comments to further improve the manuscript.

*

We thank the reviewer for this positive feedback and the helpful comments.

**

1. My main concern: causation needs to be more carefully addressed. It seems, that authors assumed that increased food disgust sensitivity may predict higher levels of fear related to COVID-19. However, higher pathogen threat (i.e. fear related to COVID-19) may influence disgust sensitivity and pathogen avoidance behavior. It is worth citing some research that demonstrates experimental manipulations of pathogen threat might up-regulate disgust sensitivity (i.e., behavioral immune system processes). The following two papers may (or may not) be useful for some of the discussion on the link between disgust sensitivity and pathogen threat during COVID-19 pandemic: https://doi.org/10.3389/fpsyg.2020.600761
https://doi.org/10.3389/fpsyg.2021.622634

The language throughout needs to be more cautious and authors should be more attentive to other potential causal scenarios.

*

We thank the reviewer for this remark. We worked through the manuscript, adapting the language, and adding more caution to the formulations. Importantly, we added more detail to the discussion:

… Still, it needs to be acknowledged that correlation does not equal causation. The pandemic comes with an increased pathogen pressure, which in turn can up-regulate disgust sensitivity (Stevenson, Saluja, & Case, 2020). For instance, Stevenson et al. (2020) found that disgust levels assessed during the time of the pandemic were higher than the values reported before the pandemic. Similarly, we noticed that the average food disgust sensitivity in our sample was higher than the values reported in previous studies conducted in Germany before the pandemic (M = 3.84 [females] and M = 3.58 [males]; Egolf et al., 2019). These findings may serve as an indication that increased pathogen threat, as imposed through the pandemic, leads to an increase in disgust sensitivity. However, these interpretations should be further underpinned by additional research.

**

2. As a general rule, a full stop is not used at the end of a title.

*

We thank the reviewer for this remark. Full stop has been removed from the title.

**

3. In the introduction, Authors write, “A common disgust response is avoidance, resulting in reduced pathogen contact and lower risk of infection (…) disgust is a pathogen-avoidance mechanism….” Consider specifying pathogen disgust here (reactiveness to contamination threats), since sexual and moral disgust do not appear to neutralize infection (see https://doi.org/10.1037/a0030778).

*

We agree with the reviewer that the pathogen disgust should be defined, especially as some of our argumentation builds on pathogen disgust (see comment below). Therefore, we have added the following definition to the introduction:

Tybur, Lieberman, and Griskevicius (2009) identified three domains of disgust: (1) pathogen disgust, (2) sexual disgust and (3) moral disgust. Pathogen disgust can be seen as reactiveness to contamination threats and motivates avoidance of infectious microorganisms, resulting in reduced pathogen contact and lower risk of infection A common disgust response is avoidance, resulting in reduced pathogen contact and lower risk of infection (Curtis, 2011; Shook, Thomas, & Ford, 2019; van Overveld, de Jong, & Peters, 2010).

**

4. Perhaps a little more justification as to why, only food disgust sensitivity was measured. More space should be devoted in the introduction and methods to describing why you chose the measures you chose (the Food Disgust Scale), and why certain related measures (e.g., the Three Domain Disgust Scale, the Disgust Scale-Revised or sets of images associated with contagion risk) were not included. More transparency about your design decisions would be helpful for readers to evaluate the study's methods.

*

We thank the reviewer for this remark. We have added the following explanations.

Introduction:

… We chose a food-specific measure of disgust sensitivity for three reasons. First, it has been demonstrated that there is a relationship between pathogen disgust and behaviours related to the pandemic (Gul et al., 2021). Given that we know that pathogen disgust and food disgust sensitivity are positively correlated (Egolf et al., 2019), we aimed to add to the evidence by investigating the role food disgust sensitivity plays in mitigating the effects of the pandemic. Second, the current pandemic has a huge impact on how we buy, prepare, and consume food. Therefore, we expect some conceptual overlap between food disgust sensitivity, hygiene, and behaviours related to the pandemic. Third, we aim to investigate whether food disgust sensitivity, due to its risk avoidance nature, explains variance beyond the domain of food. Given that enforcement of virus-mitigating measures is costly and public compliance cannot be taken for granted (Brouard et al., 2020), a better understanding of the mechanisms and motivations underlying virus-mitigating behaviours and attitudes can crucially contribute to the success of political measures aiming to contain the spread of the virus.

Methods:

The third and final section included a measure of disgust sensitivity. To the best of our knowledge, the FDS short is the only available disgust measure that specifically focuses on the domain of food. As a result, we used the 8-item short version of the Food Disgust Scale (FDS short, Hartmann & Siegrist, 2018) as a measure of food-specific disgust.

**

5. Any a priori decisions that factored into your sample size (rules of thumb, power analyses) should reported.

*

We thank the reviewer for pointing out that this information is missing. We have added it to the methods section: 

We aimed for at least 250 male and female participants each. Assuming small to medium effect sizes (d = .25), this sample size provides ample power (.80) to detect possible effects (α = .05) with a one-tailed test (Faul, Erdfelder, Lang, & Buchner, 2007). Similarly, a sample of 500 provides sufficient power to compare dependent Pearson’s correlations with common index (Faul, Erdfelder, Buchner, & Lang, 2009).

**

6. In affluent parts of the world it is possible to choose to avoid a pathogen if you think one may be present e.g. in expired food as other meals are available. In some environments people have no way to avoid similar warning signs – can it be assessed in future research directions?

*

We agree with the reviewer. We believe the environment is a very important aspect that needs to be investigated and discussed. We added it to the limitations and outlook section:

… Given that similar constructs for food disgust sensitivity have been used cross-culturally (Ammann, Egolf, Hartmann, & Siegrist, 2020; Egolf et al., 2019), another interesting endeavour would be to test whether these relationships can be found across different countries. Not only have individuals been affected differently by the pandemic depending on their socio-demographic background (Martinez, Kopp, Lalive, Pichler, & Siegenthaler, 2021), but individual disgust sensitivity also depends strongly on the environment in which an individual lives (see Ammann, Hartmann, & Siegrist, 2019 for a discussion).

---

## [Editor Report · Decision Letter 1]

1 Jul 2021

Food disgust sensitivity predicts disease-preventing behaviour beyond the food domain in the COVID-19 pandemic in Germany

PONE-D-21-12532R1

Dear Dr. Ammann,

We’re pleased to inform you that your manuscript has been judged scientifically suitable for publication and will be formally accepted for publication once it meets all outstanding technical requirements.

Kind regards,

Valerio Capraro

Academic Editor

PLOS ONE
---

## [Editor Report · Acceptance letter]

6 Jul 2021

PONE-D-21-12532R1 

Food disgust sensitivity predicts disease-preventing behaviour beyond the food domain in the COVID-19 pandemic in Germany 

Dear Dr. Ammann:

I'm pleased to inform you that your manuscript has been deemed suitable for publication in PLOS ONE. Congratulations! Your manuscript is now with our production department. 

Kind regards, 

on behalf of

Dr. Valerio Capraro 

Academic Editor

PLOS ONE